# Is Mandatory Vaccination in Population over 60 Adequate to Control the COVID-19 Pandemic in E.U.?

**DOI:** 10.3390/vaccines10020329

**Published:** 2022-02-18

**Authors:** Nikolaos P. Rachaniotis, Thomas K. Dasaklis, Filippos Fotopoulos, Michalis Chouzouris, Vana Sypsa, Antigone Lyberaki, Platon Tinios

**Affiliations:** 1Department of Industrial Management and Technology, University of Piraeus, 18534 Piraeus, Greece; 2School of Social Sciences, Hellenic Open University, 26335 Patras, Greece; dasaklis@unipi.gr; 3Department of Informatics, University of Piraeus, 18534 Piraeus, Greece; filfwt@hotmail.com; 4Department of Statistics and Insurance Science, University of Piraeus, 18534 Piraeus, Greece; mchouzouris@unipi.gr (M.C.); ptinios@unipi.gr (P.T.); 5Department of Hygiene, Epidemiology and Medical Statistics, School of Medicine, National and Kapodistrian University of Athens, 11527 Athens, Greece; vsipsa@med.uoa.gr; 6Department of Economic & Regional Development, Panteion University, 17671 Athens, Greece; alymber@panteion.gr

**Keywords:** COVID-19, mandatory vaccination, SEYLL

## Abstract

Vaccine hesitancy, which potentially leads to the refusal or delayed acceptance of COVID-19 vaccines, is considered a key driver of the increasing death toll from the pandemic in the EU. The European Commission and several member states’ governments are either planning or have already directly or indirectly announced mandatory vaccination for individuals aged over 60, the group which has repeatedly proved to be the most vulnerable. In this paper, an assessment of this strategy’s benefits is attempted by deriving a metric for the potential gains of vaccination mandates that can be used to compare EU member states. This is completed by examining the reduction in Standard Expected Years of Life Lost (SEYLL) per person for the EU population over 60 as a function of the member states’ vaccination percentage in these ages. The publicly available data and results of the second iteration of the SHARE COVID-19 survey on the acceptance of COVID-19 vaccines, conducted during the summer of 2021, are used as inputs.

## 1. Introduction

Two years after its emergence in Wuhan, China, the global spread of SARS-CoV-2, the pathogen that caused the pandemic COVID-19, continues to strain healthcare systems. By 3 January 2022, 290,157,607 confirmed cases of COVID-19, including 5,443,753 deaths, were reported globally (COVID-19 Map, Johns Hopkins Coronavirus Resource Center (www.jhu.edu, accessed on 3 January 2022)). Up to December 2020, due to the absence of effective therapies and vaccines, countries were limited to Non-Pharmaceutical Interventions (NPIs) to control the spread of the disease and minimize death rates. They included, inter alia, social distancing and lockdowns, travel restrictions, face masks, teleworking, etc. As vaccines became available, countries tried to develop exit strategies from the pandemic. These strategies aimed to balance the optimum utilization of vaccine resources that gradually became available with the retention of some NPIs.

Over the same period, the European Union (EU) reported more than 57,000,000 confirmed cases and 906,000 deaths (COVID-19 Data Explorer—Our World in Data). The increasing death toll from the pandemic in the EU was attributed to two major reasons: vaccine hesitancy, which lies behind the refusal or delayed acceptance of vaccines, and the advent of the Delta variant in March 2021, yielding higher transmissibility and mortality. In addition, there is the waning of vaccine immunity, especially for those aged over 60 who were among the first to be vaccinated. This population group accounts for more than 90% of the total deaths from COVID-19 in Europe [1].

The consequence is a vivid discussion across the EU of ways to convince the unvaccinated. With the notable exception of health care personnel, who were obliged to become vaccinated in several EU countries until November 2021, the measures progressed from positive incentives to negative incentives, most frequently in the form of COVID-19 “green passes,” and to other limitations only targeting the unvaccinated population. However, an increasing number of countries judge indirect approaches as insufficient and have implemented or considered implementing vaccination mandates (Table 1). For its own part, the European Commission has called for a horizontal EU discussion to safeguard a consistent approach, one which retains health benefits while preserving key European principles such as freedom of movement or the internal market. Some countries have defined vaccination as a legal obligation for subsections of the population, while others envisage direct penalties in the form of fines or the suspension of employment. However, even where the threat of mandatory vaccination exists, a significant fraction of the population remains unconvinced and appears to deny vaccination at all costs [2,3]. The EU discussion is followed with interest internationally; Indonesia has already mandated vaccinations, while countries from Australia to Peru, as well as US states (NY), are actively considering general or specific mandates [4].

How can the potential efficacy of vaccine mandates be judged? This question necessitates computing a counterfactual, resting on assumptions of how many more people will be vaccinated due to the mandate and who would not have been vaccinated otherwise. These assumptions need to be supplemented by a metric of the gains from vaccination, and, for this purpose, the Standard Expected Years of Life Lost (SEYLL) per person for the EU population aged over 60 years is chosen in the paper. SEYLL is a standard metric used in burden-of-disease estimations. It calculates life-years lost compared to a standardized reference life table, which is common across the EU in the cases examined. In other words, the gains of the mandates are defined as the averted SEYLL should the mandated obligation succeed—i.e., if the people who would not have been vaccinated on their own are pushed to uptake vaccines. This metric can be used to compare the potential benefits of mandatory vaccination across EU member states.

In this paper, publicly available data are used to compute the benchmark SEYLL, while findings from the second iteration of the SHARE COVID-19 survey, conducted during the summer of 2021, are used to proxy the number of people in each country who refuse voluntary vaccination. The results showcase policy recommendations regarding pandemic containment in the EU. They could support decision makers by suggesting a policy pathway to be followed until the majority of the population is vaccinated, and argue in favor or against vaccine mandates at the country or EU level.

The remainder of this paper is organized as follows: Section 2 provides a review of the relevant literature. In Section 3, the proposed methodology of the study is described, whereas in Section 4 the results are presented. Finally, Section 5 discusses the study’s results and policy implications.

## 2. Background Literature

Large differences in country-specific COVID-19 mortality rates have prompted debate and speculation about the reasons that lie behind them. Many factors have been examined recently, including genetic, viral, medical, socioeconomic and environmental ones. Nevertheless, reliable data indicated from the very beginning that the crude Case Fatality Rate (CFR) of COVID-19 is predominantly determined by the proportion of all individuals diagnosed with the SARS-CoV-2 infection who are aged over 60 [1]. In consequence, the question that naturally arises is whether the targeted vaccination of those aged over 60 would be sufficient in order to mitigate the impact of the pandemic.

In an attempt to answer this question, a stream of literature, even before the advent of vaccines, focused on the possible benefits of combining vaccinations with Non-Pharmaceutical Interventions (NPIs), assuming various levels of vaccine efficacy [5,6,7,8,9]. In order to eliminate disease transmission, a highly effective vaccine is required, and complementing vaccination with NPIs will always yield optimal containment results [10].

Vaccination against COVID-19 was initiated globally in December 2020. After a year of continuous testing, it was found that vaccine effectiveness against symptomatic disease and hospitalization, for the Delta variant that dominated until December 2021, fell after a time interval of 20–25 weeks (5 to 6 months) from vaccination completion. The waning of vaccine effectiveness is faster in older adults and those in a clinical risk group [11,12].

Vaccines aim to reduce premature deaths caused directly or indirectly by COVID-19. To capture vaccine efficiency in this paper, the metric used is SEYLL per living person aged over 60. SEYLL, a specific form of Years of Life Lost (YLL), is a standard metric used in burden-of-disease estimations. It calculates life-years lost compared to a standardized reference life table. In other words, a person’s life expectancy at each age is estimated based on the lowest observed age-specific mortality rates across all countries in the EU. The major features of this metric are:It regards all deaths as important, but those affecting younger individuals (still over 60) as particularly important, given that in their case, more life-years are lost. This has the indirect impact of placing greater weight on lower-income EU countries, in which younger deaths are more common. Thus, understanding the mortality impact of COVID-19 requires not only counting the dead but also analyzing how premature those deaths are.It values a death at a given age identically across all countries in the EU, regardless of differences in national life expectancy or income per capita. Otherwise, a death occurring in a high-longevity, richer country would count for more than one occurring in a poorer, less long-living country.

Given its egalitarian emphasis, this metric is selected in the proposed Fair Priority Model for the allocation of vaccines, thus rendering it a suitable assessment tool [13,14,15].

YLL has been used to assess COVID-19 effects in some papers, either in the study of a single country [16,17] or several countries together (e.g., [18,19,20,21]), though before vaccination was available. Only one research work took place in the first months of the vaccination campaign in Hungary [22].

This paper attempts to express SEYLL from COVID-19 per living person aged over 60 years in the EU as a function of population vaccination coverage. Developing such an aggregated model could be useful for policymakers to assess strategies that favor mandatory vaccination affecting this age group, both at the country or even the EU level, in terms of how much these strategies can reduce the total death toll should a particular vaccination level be achieved.

## 3. Methods

Available country-level and age- and gender-specific data on COVID-19 deaths occurring from 25 May 2021 to 25 November 2021 among those aged over 60 in the 27 EU countries, collected in the COVerAGE database [23] freely available from the Open Science Framework [24], were used. The COVerAGE data was processed and stratified to 3 10-year intervals, i.e., 60–69, 70–79 and 80+, separately for females and males, across countries. Countries’ missing data were filled in from their national health authority websites and national press reports. In the same 10-year intervals, age-stratified data on the weekly numbers of vaccinations in EU countries up to the 25th of November 2021 were collected from the European Centre for Disease Prevention and Control (ECDC). Germany and the Netherlands do not provide these data, so data were filled in from national press reports and relevant research work.

This specific six-months period (25 May–25 November 2021) was selected for several reasons:All EU countries had been able to provide their citizens aged over 60 the opportunity to complete their vaccination by the end of May 2021. Thus, variations observed between countries can be ascribed to individuals’ wishes to be vaccinated, rather than their ability to access the vaccine.The end-period of November 2021 coincides with the inception of discussions of progressing from disincentives imposed for non-vaccination to mandating vaccination for the general population. Vaccinations of over-60s taking place after November could plausibly be ascribed to the anticipation of future restrictions.On 26 November 2021, the WHO classified Omicron as a variant of global concern, which eventually prevailed over the Delta variant by the end of December in most EU countries. The Omicron variant is still under examination regarding its CFR and vaccines’ effectiveness against it.

The EU population aged over 60 years intending to be vaccinated was captured using data from the Survey of Health, Ageing and Retirement in Europe (SHARE) (www.share-project.org, accessed on 1 December 2021), wave 8. SHARE is a large, multidisciplinary, cross-national panel survey of micro data on the health, long-term care, socio-economic status and social and family networks of individuals aged 50 years and over. SHARE started in 2004 and was modeled based on the US HRS, whereby respondents are followed regularly after initial contact. Data are internationally comparable; translation consistency is heavily tested and checked across countries and waves [25]. SHARE has design features, such as proxy respondents, adapted for an older population; since wave 7 (2017), it covers 27 EU countries, plus Switzerland and Israel. The collection of data for SHARE wave 8 was interrupted by the pandemic in March 2020; two telephone surveys focusing on the pandemic were conducted using the SHARE panel in June 2020 and July 2021. The latter survey re-interviewed 46,989 respondents aged 50 years and older (91% of whom were 60 years and older) from the first SHARE COVID-19 Survey. It was conducted from June to August 2021 in all 28 countries participating in SHARE (26 EU countries plus Switzerland and Israel). The survey was fielded at a time when vaccine availability for the over-60 population was no longer an issue in any of the participating countries. Thus, SHARE’s second COVID-19 survey remains one of the few large-scale studies, if not the only one, that covers almost all EU countries and has collected data on individuals’ situations during the pandemic, using sampling methodologies that provide internationally comparable data.

All respondents were asked whether they had been vaccinated against COVID-19 at least once. Those who had not been vaccinated were asked “whether they had already scheduled an appointment for vaccination, wanted to become vaccinated, were still undecided or did not want to become vaccinated at all” [26]. The results of these questions are shown in Figure 1 as a percent of the 60+ population. The replies are stacked according to the extent of vaccine skepticism—starting from those already vaccinated, those with appointments, those declaring an intention and the undecided, and proceeding to those categorically declaring they will abstain. It is this last percentage—i.e., those adamant they would not be vaccinated—that is used to proxy the upper vaccination limit, in the case of fully enforced mandatory vaccinations for those aged over 60, for every EU country in SHARE. Given that the mandatory vaccination strategy is being considered predominately in the EU, the two non-EU countries, Switzerland and Israel, were excluded from the analysis. This means that a sample of 42,779 individuals over 60 years old was used.

The total expected years of life lost due to COVID-19 for the over-60 population, denoted *SEYLL_t_*, is
(1)$$SEYLL_{t}=\underset{a}{\sum }e_{a}D_{a}$$
where:

*D_a_* denotes deaths in the six-month period due to COVID-19 in a 10-year interval a, *a* = ”60–69”, “70–79”, “80+”.

*e_a_* is a measure of the expected years of life that remain to be lived for a death from any cause at age *a*. These coefficients are specified by Model Life Tables West (MLTW tables), separately for females and for males, thus forming the ‘‘standard’’ in the SEYLL metric.

In order to relate *SEYLL_t_* to the population structure of EU countries, and to capture their significantly different mortality rates, the years lost per death (*SEYLL_d_*) and the years lost per living person (*SEYLL_p_*) are considered [27]:(2)$$SEYLL_{d}=\frac{\sum_{a}e_{a}D_{a}}{\sum_{a}D_{a}}$$
(3)$$SEYLL_{p}=\frac{\sum_{a}e_{a}D_{a}}{P}$$
where *P* is the population size for people over 60. EU countries’ population structure projections for 2021 by age and gender were obtained from Eurostat (www.ec.europa.eu/eurostat, accessed on 1 December 2021).

*SEYLL_p_* is a *flow* indicator—that is, it is measured over a defined period of time. *SEYLL_p_* for the period May–November 2021 can be used as a metric to compare or rank countries’ performances using a common European benchmark. Vaccinations from that period can be taken to correspond to the effects of purely voluntary initiative; subsequent vaccinations could be due to anticipating disincentives, or even mandates.

## 4. Results

Table 2 illustrates the actual vaccination percentages by 25/11, all of which predated mandatory vaccinations, with the exception of those mentioned in the Introduction, and are hence largely voluntary. The table shows *SEYLL_d_* in years and *SEYLL_p_* in days from 25 May to 25 November for the 60+ age group in the 27 EU countries. The calculated *SEYLL_d_* values (mean = 11.3 years, s.d. = 0.74 years)—shown in Figure 2—are consistent with the results of several relevant studies, where *SEYLL_d_* was estimated to be between 10 and 13 years in developed countries [28]. The metric is useful for the purpose of comparison to other causes of death *within* EU countries, e.g., in most of them, COVID-19 is the third cause of death after cancer and cardiovascular diseases [21]. However, it does not capture the significantly different mortality rates (deaths per million people) between countries. This is obtained with the calculation of *SEYLL_p_*.

The days of life lost per living person aged over 60 for the examined 6-month period (mean = 2.64 days, s.d. = 2.42 days, min = 0.44 days, max = 9.19 days) form a representative metric of the pandemic’s significant, negative effects on every EU country. They measure how many fewer days a representative individual over 60 is expected to live as a direct result of COVID-19 morbidity occurring in these six months.

In order to relate *SEYLL_p_* for the population over 60 to their observed vaccination percentage coverage, an approximated value function was calculated from the data of Table 2 using the least-squares method. The best approximation function calculated was (Figure 3):*SEYLL_p_* = 13.423 − 0.1309 × *Vaccination percentage*
(4)

with R^2^ = 79.62%. The 95% CI for the intercept is (11.13, 15.71) (s.e. = 1.11), and for the slope is (−0.16, −0.10) (s.e.= 0.01).

The vaccination-intention percentages from the second iteration of SHARE’s COVID-19 survey correspond well with EU country-level vaccination rates reported by the European Centre for Disease Prevention and Control for the 60+ populations by 25 November 2021 (ECDC vaccinetracker, 2021). The percentage of those over 60 who say they did not want to become vaccinated is used to approximate the upper vaccination limit in the case of enforced mandatory vaccinations for these ages in every EU country (Ireland’s data were extracted from national press report and relevant research work). The actual vaccination percentages were used for the only two countries where vaccination percentage slightly exceeded this limit by 25 November 2021, namely Denmark and Portugal. Using these values as an input, estimates of the *SEYLL_p_* (in days) using the calculation approximation function are illustrated in Table 2. Figure 4 highlights the differences between the 27 EU countries’ *SEYLL_p_* for the 60+ population by 25 November 2021, and their estimated respective values in the case of mandatory vaccination. In other words, Figure 4 shows the potential gains in fewer lives lost should mandates succeed in leading every recalcitrant individual to vaccination.

## 5. Discussion

The simple calculations of this paper indicate that vaccination emerges to be the most important factor in explaining variations in EU countries’ mortality from COVID-19 for age groups over 60. This result emerges naturally, without needing to employ complex mathematical models or bring other variables to bear. Its impact is significant, not only in terms of the number of deaths but also in terms of years of life lost, which could be averted if the vaccination coverage in this age group increases.

More specifically, the values of yielded R^2^ and R^2^_adj_, 79.62% and 78.8%, respectively, indicate that almost 80% of the variability in COVID-19 mortality rates for those aged over 60 in EU countries can be explained solely by their vaccination coverage. In addition, the EU-wide approximate value function for *SEYLL_p_* yields a Mean Absolute Deviation (MAD) of 0.82 days for vaccination coverage on 25 November 2021, implying a satisfactory fit for such a parsimonious model. Therefore, although a macro-approach does not purport to explain *SEYLL_p_*_,_ which depends on a range of medical, epidemiological and socioeconomic factors of the individuals affected, the overwhelming explanation for country differences stems from the vaccination percentage. Even if, as an example, GDP per head (proxying level of development) and health expenditure in 2019 as a percentage of GDP (proxying health system readiness) are added as independent variables to a multiple regression model, with *SEYLL_p_* being the dependent variable, the explanatory power rises by a rather poor 2.5% (R^2^_adj_ = 81.29%), with only health expenditure found to be (weakly) statistically significant.

The number of deaths occurring at ages above 60 justifies policy responses to protect these vulnerable population groups. Whereas vaccine mandates exist and are common in all EU countries for children and childhood diseases, fully one year after the vaccination kick-off against COVID-19, there appears to be a reticence to proceed with equivalent mandates affecting the adult population. The need for such a mandate—in the sense of the share of the vulnerable groups who are not vaccinated voluntarily—differs widely in the EU.

However, from Figure 3, it is obvious that in, say, Bulgaria, Croatia, Cyprus, Greece, Latvia, Lithuania and Romania, even if the predicted upper vaccination limit is achieved and therefore mandates are 100% successful, *SEYLL_p_* would still remain high, as death rates are high.

This study has several limitations. As countries are at different stages in the pandemic trajectory, it is a snapshot of the impact of COVID-19 on *SEYLL_p_* by 25 November 2021. The impact of the Delta variant in the examined time span of six months is probably mostly captured, but the advent of the Omicron variant at the end of November 2021 and its effects are not. The Omicron variant spreads more easily than the Delta variant, and vaccine effectiveness against symptomatic Omicron infection is estimated to be reduced. Current vaccines are expected to protect against severe illness, hospitalization and death due to infection by the Omicron variant, but breakthrough infections in elderly people who are fully vaccinated are likely to occur, with an impact that necessitates more time to estimate.

Furthermore, *SEYLL_p_* figures may be misestimated. On the one hand, COVID-19 deaths may not be accurately recorded in some countries; most of the evidence suggests that there is a net underestimation of the total death toll on the aggregate level. On the other hand, those dying from COVID-19 may be of an at-risk population whose remaining life expectancy is shorter than the average person’s due to co-morbidities [29]. Consequently, *SEYLL_p_* due to COVID-19 may be overestimated.

The threshold of 60 years of age was used, though, of course, other thresholds could have been adopted (e.g., 65 years). It was decided to use the threshold for the definition of “older adults” (adults older than 60 years of age) indicated by the WHO’s SAGE [30]. While other factors undoubtedly affect mortality, such as race, deprivation, level of education and other medical conditions [31], age is easily verifiable and, in legal terms, could constitute a less controversial basis for vaccine mandates. In a similar fashion, age was used in all countries to prioritize vaccination rollout in the first months of 2021 [31,32].

Nevertheless, this study confirmed the large mortality impact of COVID-19 among the elderly, even when using a metric that places greater weight on deaths occurring at younger ages. It also calls for devising policies that protect vulnerable populations losing the largest number of life-years. Addressing vaccine hesitancy is challenging and requires significant efforts to increase vaccine confidence through coordinated approaches worldwide [33,34]. Finally, in order to evaluate the effectiveness of mandatory vaccination against COVID-19 for those aged over 60, comparable country-specific data on divergences in vaccination adherence must become an indispensable tool for policy discussions at the EU level—as distinct from discussions within each member state.

## Figures and Tables

**Figure 1 vaccines-10-00329-f001:**
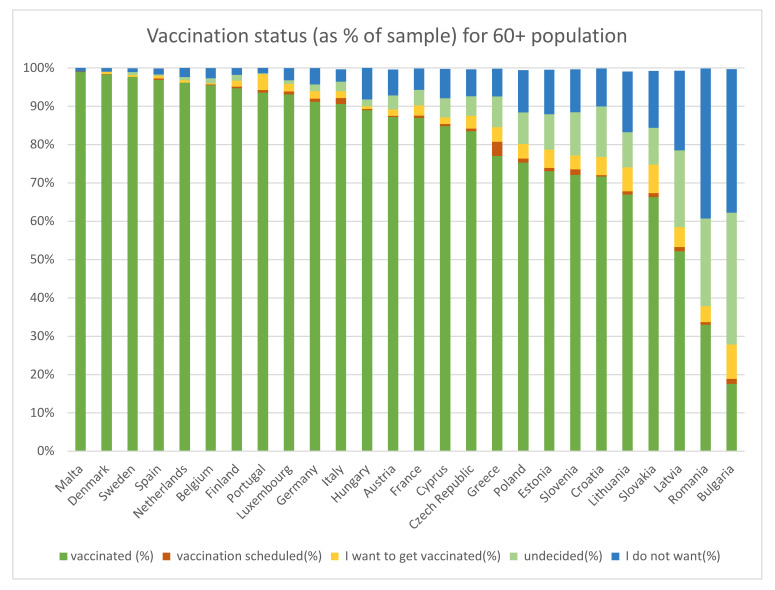
SHARE COVID-19 survey vaccination status for the 60+ sample.

**Figure 2 vaccines-10-00329-f002:**
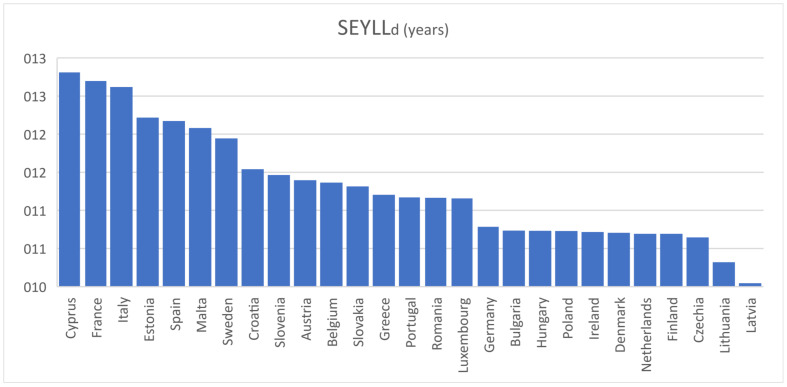
*SEYLL_d_* (in years) for the 27 EU countries.

**Figure 3 vaccines-10-00329-f003:**
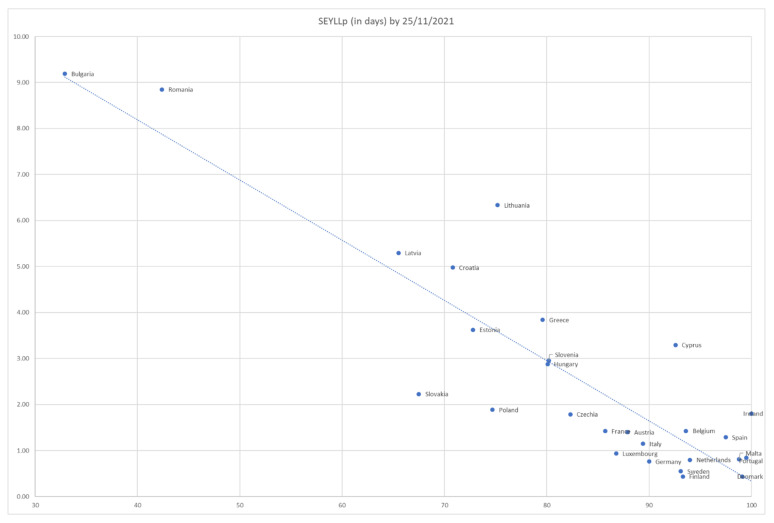
*SEYLL_p_* (in days) for the population over 60 in the 27 EU countries from 25 May 2021 to 25 November 2021 vs. their percentage vaccination coverage.

**Figure 4 vaccines-10-00329-f004:**
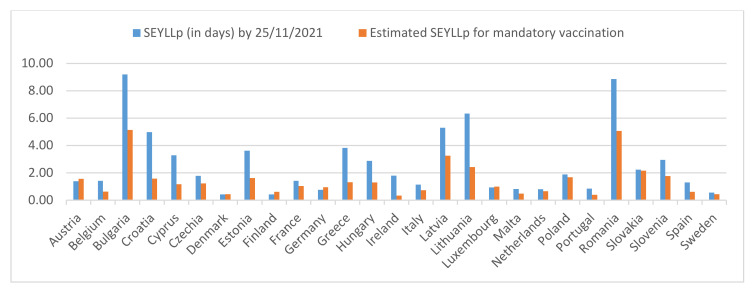
Differences between the 27 EU countries’ *SEYLL_p_* for the 60+ population by 25 November 2021, and their estimated respective values in the case of a mandatory vaccination using SHARE COVID-19 upper vaccination limits.

**Table 1 vaccines-10-00329-t001:** EU countries’ compulsory vaccination policies by 6 January 2022.

Country	Policy
Austria	Compulsory vaccination for all adults from 1 February 2022
France	Compulsory vaccination for health care personnel, firefighters and transportation workers
Greece	Compulsory vaccination for health care personnel; fines for unvaccinated people over 60 from 15 January 2022
Hungary	Compulsory vaccination for health workers, state school teachers and in public institutions
Italy	Compulsory vaccination for non-working people over 60, health care personnel, teachers, military, police and rescue crews
Latvia	Compulsory vaccination for health care personnel, teachers and care facilities

**Table 2 vaccines-10-00329-t002:** Vaccination percentages by 25 November, *SEYLL_d_* in years, *SEYLL_p_* in days from 25 May to 25 November for the 60+ age group, upper vaccination limits for 60+ age group and estimated *SEYLL_p_* for the 27 EU countries.

Country	Vaccination Percentage by 25/11 in the 60+ Age Group	*SEYLL_d_* (in Years)	*SEYLL_p_* (in Days) for the 60+ Age Group from 25/5 to 25/11	*Upper* Vaccination Limit from SHARE COVID-19 (%) for Ages 60+	*SEYLL_p_* Estimation (Days) for the Upper Vaccination Limit
Austria	87.9	11.40	1.40	90.51	1.58
Belgium	93.6	11.37	1.43	97.65	0.64
Bulgaria	32.9	10.74	9.19	63.30	5.14
Croatia	70.8	11.54	4.98	90.43	1.59
Cyprus	92.6	12.81	3.29	93.60	1.17
Czechia	82.3	10.65	1.79	93.11	1.24
Denmark	99.1	10.71	0.44	99.10 *	0.45
Estonia	72.8	12.22	3.62	90.07	1.63
Finland	93.3	10.70	0.44	97.84	0.62
France	85.7	12.70	1.42	94.62	1.04
Germany	90	10.79	0.76	95.29	0.95
Greece	79.6	11.21	3.84	92.38	1.33
Hungary	80.1	10.73	2.88	92.67	1.29
Ireland	100	11.86	1.80	100.00	0.33
Italy	89.4	10.72	1.15	96.93	0.74
Latvia	65.5	12.62	5.30	77.69	3.25
Lithuania	75.2	10.05	6.34	84.00	2.43
Luxembourg	86.8	10.32	0.93	94.96	0.99
Malta	98.8	11.16	0.81	98.80	0.49
Netherlands	94	12.08	0.80	97.43	0.67
Poland	74.7	10.70	1.88	89.67	1.68
Portugal	99.5	11.80	0.84	99.50 *	0.40
Romania	42.4	10.73	8.85	63.81	5.07
Slovakia	67.5	11.17	2.23	86.02	2.16
Slovenia	80.2	11.17	2.95	88.97	1.78
Spain	97.5	11.32	1.29	97.82	0.62
Sweden	93.1	11.47	0.55	99.17	0.44

* Actual vaccination percentages by 25 November 2021.

## Data Availability

Not applicable.

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
