# Peer review of "Is Mandatory Vaccination in Population over 60 Adequate to Control the COVID-19 Pandemic in E.U.?"

_vaccines, 2022, doi:10.3390/vaccines10020329_

Round 1

Reviewer 1 Report

This is a very useful paper in pointing the value of vaccination in adults over the age of 60.  The presence or absence of vaccination accounts for a very large proportion of the variation in the SEYLL measure.

I have a few suggestions.

  1. I know that many countries use the age of 60 as a threshold.  But in some respects, it is arbitrary.  It reflects something of a folk conception of what constitutes "old" or  "gradually losing immune function." How would using a different age threshold affect the results?  This is especially important these days, when 60, societally, is sometimes called the "new 50."  
  2. I also think that age is easy to mark but over-relied upon.  Socioeconomic circumstances have been found to be highly related to outcomes. Can you work from home? Are you living in crowded conditions? Can you afford medical care?  There is nothing wrong with relying on age as a marker, but it may lead us to ignore other markers and perhaps more important ones for better or worse outcomes.
  3. I think the paper would be more persuasive if it used one or two other dependent measures as well and showed that the results are not merely a function of the particular dependent measure chosen. If the results replicated across other dependent measures, they would be more persuasive I think to those in decision-making capacities.
  4. At some level, we knew that vaccination matters for people over 60.  The question that causes more of a struggle in practice is one of how you actually convince people to get vaccinated.
  5. I wonder about the durability over time of such results.  COVID has proved to be very clever in outwitting us.  Omicron is already not so responsive to the old vaccines, and the next variant may be less responsive.  Suppose the disease keeps mutating, which it seems to be doing.  Won't analyses such as these always be one step beyond the virus?  That is, the results could become out of date as fast as the vaccines do, and seem to be becoming.
  6. There might be something to be said for a more complex analysis that takes into account other variables such as SES and perhaps sheer availability of vaccines. In some places, people just can't get them, so then what?
  7. I think this paper is publishable but it would benefit from being somewhat broader in its focus.

Reviewer 2 Report

First of all, I would like to thank for the opportunity to review this paper. COVID-19 is an ongoing pandemic that has resulted in global health, economic and social crises. Actually, the vaccination campaign is the first method to counteract the COVID-19 pandemic; however, sufficient vaccination coverage is conditioned by the people’s acceptance of these vaccines in the general population and health care workers. In this context, the paper under review is aimed at assessing the mandatori vaccination strategy benefits among over60 in E.U.

The subject under study is certainly very important, especially in the historical period we are experiencing. The article presents interesting results but, but it is nevertheless believed that, given the organization of the contents and the description of the same, the manuscript can be improved. I would like to encourage authors to consider several issues to be improved.

Introduction: The authors should make clearer what is the gap in the literature that is filled with this study? The authors do not frame their study within the vast body of literature that addressed the coverage and acceptance of the vaccination in the adult population (refer to articles with DOI: https://doi.org/10.3390/vaccines9111222). Before thinking to a compulsory strategies, what are the evidence related to COVID-19 vaccine acceptance in Europe?

Methods: Part of the survey was conducted by a survey, but very little is described in methods section with regard to the sample, the tool used and the reference population.

The survey was conducted using a standard questionnaire? The use of an unreliable instrument is a serious and irreversible limitation of the study. Moreover, no mention to a validation process is reported. What about face validity, reliability and intelligibility?

The enrolment procedure must be better specified. How was the sample selected? What is the reference population? what is the minimum sample size?

Ethical Issue: although financed by a EU project an Institutional Review Board approval must obtained, since the study deals with sensible data (although anonymous). A reference number should be reported.

Discussion: I also suggest expanding. What is the possible international contribution of the study to the literature? What are the implications of the study? Emphasize the contribution of the study to the literature. The discussion must be updated with one of the principal debated argument in this epidemiological context: the use of a green pass linked to vaccination practice (refer to articles with DOI: https://doi.org/10.3390/vaccines9111222).

Author Response

1. Introduction: The authors should make clearer what is the gap in the literature that is filled with this study? The authors do not frame their study within the vast body of literature that addressed the coverage and acceptance of the vaccination in the adult population (refer to articles with DOI: https://doi.org/10.3390/vaccines9111222). Before thinking to a compulsory strategies, what are the evidence related to COVID-19 vaccine acceptance in Europe?

- Thank you for the useful comment. We have added the reference in the revised manuscript and we have tried to pitch it in the context of needing a European metric for judging the cost/benefit balance of mandatory vaccines (lines 58-60 and 65-76 in the Introduction section).  The literature gap is pinpointed in the Background Literature section (lines 125-134) in the revised manuscript.

2. Methods: Part of the survey was conducted by a survey, but very little is described in methods section with regard to the sample, the tool used and the reference population. The survey was conducted using a standard questionnaire? The use of an unreliable instrument is a serious and irreversible limitation of the study. Moreover, no mention to a validation process is reported. What about face validity, reliability and intelligibility? The enrolment procedure must be better specified. How was the sample selected? What is the reference population? what is the minimum sample size?

- An informative passage on SHARE (lines 162-194) plus Figure 1 with the replies to the vaccination intention answer of the survey are added in the Methods section of the revised manuscript.

3. Ethical Issue: although financed by a EU project an Institutional Review Board approval must obtained, since the study deals with sensible data (although anonymous). A reference number should be reported.

- The SHARE study is subject to continuous ethics review. Wave 8 of SHARE and the continuation of the project were reviewed and approved by the Ethics Council of the Max Planck Society (29.5.2020).

4. Discussion: I also suggest expanding. What is the possible international contribution of the study to the literature? What are the implications of the study? Emphasize the contribution of the study to the literature. The discussion must be updated with one of the principal debated argument in this epidemiological context: the use of a green pass linked to vaccination practice (refer to articles with DOI: https://doi.org/10.3390/vaccines9111222).

- As stated in the Introduction Section of the revised manuscript (lines 60-63), there are implications of the study beyond E.U. The green pass use is discussed in lines 48-51. For the policy implications please refer to the Discussion Section of the revised manuscript, lines 295-298 and 323-329.

Round 2

Reviewer 2 Report

The paper was improved and it is now suitable for publication